



# Retrieval of the total precipitable water vapor and cloud liquid water path over ocean from the Feng-Yun 3D microwave temperature and humidity sounders

Jun Yang[1,*], Fuzhong Weng[1], Hao Hu[1], Peiming Dong[1]

[1] State Key Laboratory of Severe Weather, Chinese Academy of Meteorological Sciences, Beijing 100081, China

*Correspondence to*: Jun Yang (yangjun@cma.gov.cn)

**Abstract.** Feng-Yun 3D (FY-3D) satellite is the latest polar-orbiting meteorological satellite launched by China and carry 10 instruments onboard. Its microwave temperature sounder (MWTS) and microwave humidity sounder (MWHS) can acquire a total of 28 channels of brightness temperatures, providing rich information for profiling atmospheric temperature and moisture. However, due to a lack of two important frequencies at 23.8 and 31.4 GHz, it is difficult to retrieve the total precipitable water vapor (TPW) and cloud liquid water path (CLW) from FY-3D microwave sounder data as commonly done for other microwave sounding instruments. Using the channel similarity between Suomi NPP advanced technology microwave sounder (ATMS) and FY-3D microwave temperature and humidity sounders, a machine learning technique is used to generate the two missing low frequency channels of MWTS and MWHS. Then, a new data set named as a combined microwave sounder (CMWS) is obtained and has the same channel setting as ATMS but the spatial resolution is consistent with MWTS. It is shown that the mean absolute errors of the two simulated channels are both between 3 and 4 K. The simulation errors mainly distribute in the high latitude regions, coastlines and the boundaries of some heavy rainfall. A statistical inversion method is adopted to retrieve TPW and CLW over oceans from the FY-3D CMWS. The inter-comparison between different satellites shows that the inversion products of FY-3D CMWS and Suomi NPP ATMS have good consistency in magnitude and distribution.

## 1. Introduction

Clouds and water vapor are vital for regulating the global radiation budget, and their distribution and evolution in the atmosphere play an extremely important role in global weather and climate change (Stephens, 2005; Yang et al., 2017; Yang et al., 2018). The total precipitable water vapor (TPW) and cloud liquid water path (CLW) are two key physical quantities for assessing cloud water resources in the air. The radiosonde observation



(RAOB) can obtain direct measurement of water vapor, but the operational RAOB measurements are only performed twice a day at several hundred fixed stations around the world, which makes its temporal and spatial

resolution have great limitations (Gui et al., 2017). Therefore, remote sensing inversion has become an important means to obtain TPW and CLW. Both ground-based and satellite retrieval methods have been widely developed in the microwave range to get local or global cloud water content. Ground-based microwave radiometer can obtain continuous TPW and CLW measurements with a very high temporal resolution for a stationary observation site (Lin et al., 2001; Cadeddu et al., 2009; Cimini et al., 2010; Yang and Min, 2018).

The limitation of ground observation lies in its poor spatial distribution, especially on the ocean.

Compared with ground-based observation, satellite measurement can provide TPW and CLW inversion products on a global scale although its temporal resolution is relatively low. The large-scale TPW and CLW distribution obtained by satellite inversion, in addition to being used for global cloud water resource assessment, another important function is to use as cloud detection criterion for quality control in satellite data assimilation.

The brightness temperature measured by satellites is affected by many factors such as surface emissivity, water vapor and cloud liquid water content. In the ocean, the surface emissivity is relatively uniform. Normally, TPW and CLW can be retrieved using two microwave channel measurements. In the early days of satellite applications, Grody et al. (1980) applied the data from 21 and 31 GHz channels of scanning microwave spectrometer mounted on Nimbus 6 to retrieve CLW over the Pacific Ocean. Prabhakara et al. (1983) developed

a method to retrieve CLW from the data on Nimbus 7 scanning multichannel microwave radiometer at 6.6 and 10.7 GHz. With the launch of special sensor microwave/imager (SSM/I) aboard the defense meteorological satellite program, a large number of satellite inversion algorithms based on this passive microwave sensor were reported to obtain globe distribution of TPW and CLW over ocean (Greenwald et al., 1993; Liu and Curry, 1993; Weng and Grody, 1994; Ferraro et al., 1996). There are four frequencies (at 19.35, 22.235, 37 and 85.5

GHz) in SSM/I, and the existing algorithms mainly use the combination of 22.235GHz and another frequency to invert TPW and CLW. The successive launch of advanced microwave sounding unit and advanced technology microwave sounder (ATMS) further promoted the application of satellite inversion of TPW and CLW over ocean (Dyras and Serafin-Rek, 2002; Mo and Liu, 2008; Weng et al., 2012). Both statistical (Grody et al., 2001) and physical inversion (Weng et al., 2003) methods had been proposed to improve the inversion

accuracy of TPW and CLW using two window channels (at 23.8 and 31.4 GHz, respectively).



Feng-Yun 3D (FY-3D) satellite, which was launched in November 2017, is the fourth satellite of China's second-generation polar-orbiting satellite. There are 10 sets of advanced remote sensing instruments on the FY-3D, among which microwave temperature sounder (MWTS) and microwave humidity sounder (MWHS) are mainly designed to obtain atmospheric temperature and water vapor profiles. The MWTS operates in an oxygen

absorption band of 50-60 GHz and is subdivided into 13 channels. There are total 15 channels in MWHS, which are located in oxygen absorption zone (8 channels near 118.75 GHz), water vapor absorption zone (5 channels near 183.31 GHz) and window zone (at 89 and 150 GHz), respectively. Using the dual oxygen absorption bands in MWTS and MWHS, several pairs of oxygen channels can be applied to compute cloud emission and scattering index at different height levels (Han et al., 2015), where each pair of oxygen channels has a similar

peak weighting function height. The strong water vapor absorption channels at high frequency in MWHS are very sensitive to small changes of water vapor, and it is only suitable for inversion of TPW below 7 kg/m$^2$ (Melsheimer and Heygster, 2008). Therefore, it is often used for water vapor retrieval in polar regions. Two window channels in MWHS can be used to retrieve cloud scattering index (Bennartz et al., 2002) and cloud ice water path (Zhao and Weng, 2002). Although some cloud water information can be obtained using different

algorithms based on the existing microwave channels of FY-3D, it is still very difficult to give specific values of TPW and CLW. The main reason is the lack of two important window channels, 23.8 and 31.4 GHz, in FY-3D. Fortunately, compared with ATMS aboard Suomi NPP satellite, MWTS and MWHS contain all ATMS channel information except two low-frequency window channel and thus, ATMS data provide a good opportunity for simulating FY-3D MWTS measurements at any frequencies.

In this paper, we introduced a channel simulation algorithm from the machine learning (ML) to generate these two window channels, which makes it possible to retrieve TPW and CLW from FY-3D microwave sensors. In the following section, the channel simulation method based on ML is first discussed. Some ML simulation results and accuracy assessments are presented in section 3. In section 4, the inversion results of TPW and CLW based on FY-3D are introduced and compared with those of ATMS. The brief conclusions and

recommendations for future work are discussed in section 5.

## 2. Channel simulation algorithm based on machine learning

For any field of view (FOV), the measurements of each channel on the same satellite sensor should have a certain correlation, because their corresponding surface types and atmospheric environment are exactly the



same. Therefore, we can establish the relationship between two low-frequency window channels and other

channels of ATMS through the ML model for all FOVs. Since MWTS and MWHS contain all channel settings in ATMS except for the two low-frequency window channels, we can match FY-3D data to ATMS level by cross calibration, thus realizing the prediction of missing channel values in FY-3D using ATMS training model. The flow chart of the proposed channel simulation algorithm is shown in Figure 1, which includes three main steps: footprint matching, cross calibration and machine learning. Firstly, each channel of MWHS transforms

to a new channel consistent with the spatial resolution of MWTS by performing footprint matching with MWTS. Then we can get a 28-channel data set, 20 of which have the same channel properties as ATMS. Secondly, these 20 channels will be cross-calibrated with the corresponding channels in ATMS. Thirdly, the typical ATMS samples are trained using ML method, and the model relationship between other channels and two low-frequency window channels in ATMS can be established. Next, the corresponding 20 channels in FY-3D are

input into the model to obtain two simulated low-frequency channels. It will combine 20 channels in FY-3D to form a new data set, named combined microwave sounder (CMWS), which is identical to the channel settings of the ATMS, but has the same observation range and resolution as the MWTS. Several key steps will be introduced in the following subsections.

### 2.1 Footprint matching

Although both MWTS and MWHS are mounted on the FY-3D satellite, they are two completely independent sensors, which also results in the difference of FOV size and FOV number on each scan line between MWTS and MWHS. For example, there are 98 FOVs on each scan line in MWHS, while only 90 FOVs in MWTS. In order to make the pixels corresponding to MWTS and MWHS have the same observation position and instantaneous FOV, we need to perform footprint matching for the original observation data. Since the spatial

resolution of MWHS is higher than that of MWTS, an alternative method is to use B-G method (Backus and Gilbert, 1968) to re-sample MWHS to produce observations with the same resolution as MWTS. However, on the FY-3D satellite, the scan times of MWTS and MWHS does not match well, which makes the brightness temperatures observed by them do not overlap well in space. Therefore, we adopted a simple weighted average to match the MWHS observations to the MWTS resolution level. By setting a distance threshold, for each FOV

in the MWTS, find all points below the threshold in the MWHS, and calculate the average of these points to obtain the matched MWHS brightness temperature values. Figure 2 shows two typical MWHS channels (89 and 183.31±1.0 GHz) on July 9, 2018 and their results after footprint matching with MWTS. The left column


represents the original measurements of MWHS, and the right column is the results after matched. The matched

results are very close to the original observations in both intensity and distribution of brightness temperature,

which ensures that our footprint matching does not lose too much information.

## 2.2 Cross calibration

Although the channel settings between ATMS and CMWS are basically the same, the brightness temperatures

of ATMS and CMWS come from different satellite measurements, which inevitably leads to a certain deviation

between the two observations due to the differences in hardware processing and radiometric calibration. Since

our training samples are all from ATMS, and the input data of the model prediction comes from CMWS, it

becomes indispensable to perform a cross calibration process, which will ensure the measured values on each

channel between the two instruments are as consistent as possible.

The sub-satellite trajectories of Suomi NPP and FY-3D satellites are very close each other on February 1-2,

2018, which allows us to use the data of these two days to cross-calibrate ATMS and CMWS. For each FOV in

CMWS, the arc length and observation time difference between the pixel and all observation points in ATMS

are calculated. The pair of FOV with the shortest distance may be the observation of the same point by two

sensors. By setting a distance threshold and a time threshold, all matching pairs of FOVs satisfying the threshold

conditions can be considered as the same observation point and used for cross calibration. A simple linear

regression method is adopted to derive a linear regression equation for each channel, which can match the

observations of CMWS to the level of ATMS. Table 1 provides the linear fitting coefficients (slope and intercept)

and mean absolute error for each channel.

## 2.3 Machine learning

After completing the footprint matching of MWTS and MWHS and the cross calibration between ATMS and

CMWS, the next key step of channel simulation is the application of ML algorithm. ML can build an effective

training model by learning finite samples, which can not only predict the known data but also the unknown data.

ML can be divided into supervised learning and unsupervised learning. Under supervised learning, each group

of training data has a clear mark or result. In unsupervised learning, data is not specifically identified, and

learning model is designed to infer some of the intrinsic structure of data. The issue to be solved in this paper

is a regression problem, so supervised learning should be adopted to train the model. Traditional supervised

learning algorithms (Russell and Norvig, 2010) include k-nearest neighbor, neural network, support vector



machine, decision tree, etc. Recently, the ensemble algorithms have been developed, which solve the single prediction problem by establishing a combination of several models (Rokach, 2010). It works by generating multiple classifiers/models, each of which learns and makes predictions independently. These predictions are ultimately combined into a single prediction, which is usually better than any single estimator.

Random forest (RF) is a ML algorithm that integrates multiple trees by the idea of ensemble learning (Breiman, 2001). Its basic unit is the decision tree, and each tree is bootstrapping random sampling from the training set. There is no correlation between each decision tree in a RF model. The basic schematic diagram of the RF is shown in Figure 3. A decision tree consists of a root node, internal nodes and leaf nodes. First, the root node splits according to randomly selected data set attributes, and then similar splitting process is repeated in the

internal nodes until the node satisfies the maximum depth or reaches the maximum number of leaf nodes. Each leaf node returns a predicted value. The predicted result of a decision tree can be acquired by averaging the predicted values of all leaf nodes on the tree. Finally, the results of each decision tree are averaged to improve the ultimate predictive accuracy and control over-fitting. RF has the following main advantages: 1) Training can be highly parallelized, which makes training speed of the model have great advantages, especially for large

samples such as satellite data; 2) Because the decision tree node splitting feature can be selected randomly, the training model can still be efficient even if the dimension of sample features is very high; 3) The training model has small variance and strong generalization ability because of random sampling.

## 3. Accuracy evaluation and channel simulation examples

In ATMS, there are remaining 20 channels besides two low-frequency window channels. To determine whether

all 20 channels should be involved in the training of the RF model, we conducted some sensitivity tests based on a very popular ML software package scikit-learn (https://scikit-learn.org/stable/index.html). In order to facilitate the pixel-by-pixel quantitative evaluation of the simulation results, the data in the training set and the test set are all from ATMS. The full-day data of February 1, 2018 was chosen as the training set, and the data of February 2, 2018 was adopted as the test sample. By comparing the simulation results with the actual

observations, the mean absolute error (MAE) of the two low-frequency window channels (represented by Ch1 and Ch2 respectively) in different channel combinations and different model parameters can be obtained. The simulated error of each single channel is displayed in the left panel of Figure 4. It is clear that MAEs of channels 3, 4, 5 and 16 are significantly lower than those of other channels. Considering that the weighting functions and





peak heights of these four channels are very similar to those of Ch1 and Ch2 (see the Figure 2 in Weng et al.,

2012), it has natural advantages to simulate Ch1 and Ch2 using these four channels. In addition to these four

channels, other channels in ATMS are added to the training model one by one to test their effects on the

simulation accuracy of Ch1 and Ch2. The right panel of Figure 4 denotes different MAEs when each channel

is progressively added to the training model. The trend of simulation accuracy for Ch1 and Ch2 is basically the

same. When the channel used in the model increases from channel 6 to channel 11, the MAE is decreasing, and

from channel 12 to channel 15, the MAE is starting to increase again. Channel 17 to channel 22 are high

frequency channels in ATMS. When channel 17 and channel 18 are added to the training model, the simulation

errors of Ch1 and Ch2 can be further reduced, but the introduction of channel 19 to channel 22 makes the MAEs

of Ch1 and Ch2 slightly increased again. These sensitivity tests show that channel 12 to channel 15 and channel

19 to channel 22 cannot play a positive role in the simulation of Ch1 and Ch2. Therefore, in the next analysis,

we totally adopted 12 channels to simulate the missing channels at frequencies 23.8 and 31.4 GHz in FY-3D.

In a RF model, in order to obtain the highest prediction accuracy and reduce the computation time, it is necessary

to optimize some input parameters. We did some sensitivity tests to determine the values of these parameters

(see Figure 5). The more decision trees are used in the RF model, the higher the accuracy of the model is, but

the number of trees seriously affects the training speed. It is a reasonable choice to adopt 130 decision trees in

our RF model. The maximum depth of each tree determines the extent of tree expansion. More depth does not

significantly increase the accuracy of the model, so we set the maximum depth to 30. The number of features

to consider when looking for the best split obviously affects the accuracy of the model, and the simulation error

is the smallest when all 12 channels are used. The other parameters, such as the maximum number of leaf nodes,

the minimum number of samples required to split an internal node, the minimum number of samples required

to be at a leaf node, can obtain the best simulation accuracy by directly using the default values of the software

package scikit-learn.

Using the training channels and ML parameters determined by the aforementioned sensitivity tests, the two

ATMS low frequency window channels on July 10, 2018 were simulated. In order to increase the

representativeness of training samples, our training set consists of two categories; one is to select one-day data

from each month in the past year, which mainly represents the basic global climate trends in different seasons.

The other type is the data of the day before the forecast sample, which mainly on behalf of the global

approaching weather system. The point-to-point accuracy evaluation results for Ch1 simulation are shown in



Figure 6. The top row of Figure 6 represents the original ATMS measurement and ML simulated result for Ch1 on July 10, 2018. The bottom row of Figure 6 shows the difference between observation and simulation, as well

as the scatter density of observed and simulated brightness temperatures. The similar accuracy evaluation images for Ch2 simulation are depicted in Figure 7. Visually, the range of error distribution of Ch1 and Ch2 simulation is very similar, which mainly distributed in the high latitude regions, coastlines and the edge of some heavy rainfall. It should be noted that the MAE of Ch1 simulation is slightly bigger than that of Ch2 simulation, corresponding to 3.5252K and 3.4151K respectively.

A quantitative assessment of the simulation results of the two channels based on ATMS has been completed previously. We can get Ch1 and Ch2 simulation results based on FY-3D observation data by replacing the predictive input of ML model with FY-3D measurements. Figure 8 shows the simulation results of the two low-frequency window channels based on FY-3D observations on September 13, 2018. The left column is the actual observations of ATMS, and the right column is the simulation results based on FY-3D observations. The upper

row is the observation and simulation of 23.8 GHz and the lower row is the related results of 31.4 GHz. It can be seen that the simulation results of the two channels are very similar to the observations of the ATMS, especially on the ocean. On September 13, 2018, the super typhoon "Mangkhut" (about at $15°N, 130°E$) is developing vigorously in the Northwest Pacific Ocean. It is clear that our simulation results for both Ch1 and Ch2 can give the accurate locations and basic form of the typhoon. It should be pointed out that the results of

the simulation using FY-3D will definitely be lower than the accuracy of quantitative evaluation by ATMS. This is because the cross calibration between ATMS and FY-3D will inevitably introduce some new errors. Typically, the simulation error of the north and south poles is relatively large, and the simulated brightness temperature of the typhoon area is still a little weak.

## 4. Retrieval of the total precipitable water vapor and cloud liquid water path

Traditionally, retrieving TPW and CLW from satellite measurements is mainly aimed at the ocean surface because of its relatively low and uniform surface emissivity. Although several methods have been reported to retrieve TPW from land surface (Aires et al., 2001; Boukabara et al., 2010; Zhou et al., 2016), it is still very difficult to retrieve TPW and CLW from non-oceanic surface because of its pixel-by-pixel high uncertainty of surface emissivity. The purpose of this paper is not to propose a new inversion method, mainly to test the

feasibility of TPW and CLW inversion based on FY-3D microwave observation data. Therefore, the inversion



of TPW and CLW mainly focuses on the ocean. After the high-precision simulation of 23.8 and 31.4 GHz

channels based on FY-3D measurements, the inversion of TPW and CLW over ocean becomes relatively easy.

There are two inversion methods over ocean widely used in the field of microwave remote sensing, one is

Grody's statistical inversion method (Grody et al., 2001) and the other is Weng's physical inversion method

(Weng et al., 2003). In theory, physical inversion method should have higher retrieval accuracy than statistical

inversion method. However, in Weng's physical inversion model (Weng et al., 2003), some precise ocean and

cloud parameters need to be input, such as sea surface temperature, wind speed and cloud layer temperature.

These parameters are usually difficult to obtain precise measurements. An alternative is to get them from global

forecast system or from reanalysis data. Considering the RMS error for non-precipitation CLW retrieval is no

more than 0.05 mm (Grody et al., 2001), we select Grody's statistical method to retrieve TPW and CLW over

ocean from FY-3D in this paper. The equations for statistical inversion can be expressed as,

$$\text{TPW} = \cos\theta \left[ a_0 + a_1 \ln(T_s - T_{b23}) + a_2 \ln(T_s - T_{b31}) \right] \tag{1}$$

$$\text{CLW} = \cos\theta \left[ b_0 + b_1 \ln(T_s - T_{b23}) + b_2 \ln(T_s - T_{b31}) \right] \tag{2}$$

where θ is the local zenith angle, $T_s$ is the surface temperature, $T_{b23}$ and $T_{b31}$ represent the observed

brightness temperatures of 23.8 and 31.4 GHz, respectively. To ensure $T_s$ is larger than $T_{b23}$ and $T_{b31}$ over

ocean, the value of $T_s$ is set to 285 K. The coefficients $a_0, a_1, a_2, b_0, b_1, and\ b_2$ can be obtained by

performing regression analysis on the simulated channel measurements. The specific coefficients are as follows,

$$a_0 = 247.92 - (69.235 - 44.177 \cos\theta) \cos\theta \tag{3}$$

$$a_1 = -116.27, a_2 = 73.409 \tag{4}$$

$$b_0 = 8.240 - (2.622 - 1.846 \cos\theta) \cos\theta \tag{5}$$

$$b_1 = 0.754, b_2 = -2.265 \tag{6}$$

Using the above formulas and combining the brightness temperature values of 23.8 and 31.4 GHz channels

simulated in CMWS, the specific values of TPW and CLW on the ocean can be obtained efficiently. The

validation of TPW retrieved from satellite can be carried out by various methods. On the one hand, it can be

compared with the direct measurements of RAOB or the inversion results of ground-based microwave

radiometer; on the other hand, it can also be done by inter-comparing with the inversion product of other

satellites. The accuracy verification of the CLW is mainly compared with the retrieval results of the inversion



of ground-based microwave radiometer or the retrieval product of other satellites. Here, we mainly verify the inversion effect of TPW and CLW by comparing with ATMS inversion results. Figure 9 and Figure 10 show

the inversion results on July 8, 2018 and September 11, 2018, respectively. Each of these two days has a super typhoon developing in the Northwest Pacific Ocean, corresponding to the typhoon "Maria" and "Mangkhut". To better increase the diversity of data, Figure 9 shows the inversion results of the descending orbit, and Figure 10 denotes the results of the ascending orbit. From the comparison of two examples, the inversion results of TPW and CLW based on FY-3D CMWS are very similar to those of ATMS inversion both in intensity and

distribution.

## 5. Conclusion and discussion

FY-3D satellite is the latest polar-orbiting meteorological satellite launched by China, which carries 10 sets of advanced monitor instruments. The MWTS and MWHS mounted on the FY-3D provide a total of 28 channel observations that greatly improve the inversion accuracy of temperature and humidity profiles. In particular,

the 50-60 GHz oxygen channels in the MWTS and the oxygen channels near the 118.75 GHz in the MWHS can be used not only for mutual backup but also for retrieving of clouds and precipitation using dual oxygen absorption channels. However, due to the lack of two important low-frequency window channels, at frequencies 23.8 and 31.4 GHz, it is hard to retrieve TPW and CLW using FY-3D microwave measurements. The similarity of channel settings between NPP ATMS and FY-3D MWTS&MWHS provides a good opportunity for us to

simulate these two missing channels. Firstly, based on the correlation between measurements of all channels on the same sensor, the model relationship between other channels and two low-frequency window channels in ATMS is established by ML algorithm. Then cross calibration between MWTS and MWHS measurements and corresponding ATMS channels is carried out. Finally, the calibrated FY-3D measurements are input into the training model to obtain the simulated values. The results of quantitative evaluation show that the MAEs of the

two channels are both between 3 and 4 K, and the maximum error mainly occurs in high latitude regions, coastlines and the vicinity of some heavy rainfall. Especially in the central region of typhoon, the simulated brightness temperatures are somewhat lower than those ATMS measurements. It should be pointed out that although the simulation errors of the two missing channels still seem to be a little large, the simulation accuracy is sufficient to meet the quality control requirements of satellite data assimilation and atmospheric profile

parameter inversion. In the next step, we will build training models for ocean and land respectively, so as to

further improve the accuracy of channel simulation.

After simulated the 23.8 and 31.4 GHz channels using FY-3D microwave observations, Grody's statistical

method was adopted to reverse TPW and CLW over ocean. By comparing with different satellite inversion

products, it can be found that the inversion results of TPW and CLW based on FY-3D are in good agreement

with those of ATMS in both strength and distribution. Actually, the channel settings of CMWS data set

established by machine learning are basically consistent with ATMS, which makes the existing inversion

algorithms based on ATMS can be seamlessly transplanted into CMWS. In addition, after the successful

simulation of 23.8 and 31.4 GHz channels, we can even compose a new CMWS data set of 30 channels (plus

13 channels of MWTS and 15 channels of MWHS), which will provide more details for the inversion of

temperature and humidity profiles in the vertical direction. Thus, the new CMWS data set has great application

prospects, especially in the inversion of typhoon warm core structure, typhoon location and intensity

determination, and precipitation estimation. At the same time, the next FY-3 satellite FY-3E will soon be

launched, in which two window channels (at 23.8 and 31.4GHz) will be added in the new microwave sounder.

The simulation method proposed in this paper can also provide a good proxy simulation for FY-3E.

***Code and Data availability.*** Data used in this study can be made available upon request to the author.

***Author Contributions.*** JY performed the calculus, designed and conducted the experiments and wrote the paper;

FZW edited and supervised the form of the paper; HH validated the results; PMD provided help for coding

design and modified the paper.

***Competing interests.*** The authors declare that they have no conflict of interest.

***Acknowledgements.*** This work is partly supported by the National Key Research and Development Program of

China (2018YFC1506500) and the National Natural Science Foundation of China (41675030). We also

gratefully acknowledge the support from National Satellite Meteorological Centre under agreement FY3 (02P)-

MAS-1803.



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



**Figure Captions**

Figure 1. Flow chart of channel simulation algorithm based on machine learning.

Figure 2. Footprint matching examples for two typical channels.

Figure 3. Schematic diagram of random forest.

Figure 4. Simulation errors in different channel combinations.

Figure 5. Simulation errors under different RF model parameters.

Figure 6. Accuracy evaluation for Ch1 simulation.

Figure 7. Accuracy evaluation for Ch2 simulation.

Figure 8. Compared between the simulation and observation.

Figure 9. Comparison of retrieved TPW (top row) and CLW (bottom row) between ATMS (left column) and

CMWS (right column) for the descending orbit measurement on July 8, 2018.

Figure 10. Comparison of retrieved TPW (top row) and CLW (bottom row) between ATMS (left column) and

CMWS (right column) for the ascending orbit measurement on September 11, 2018.

Table 1. Slope, intercept and mean absolute error between corrected CMWS (as x values) and ATMS (as y

values) brightness temperatures.



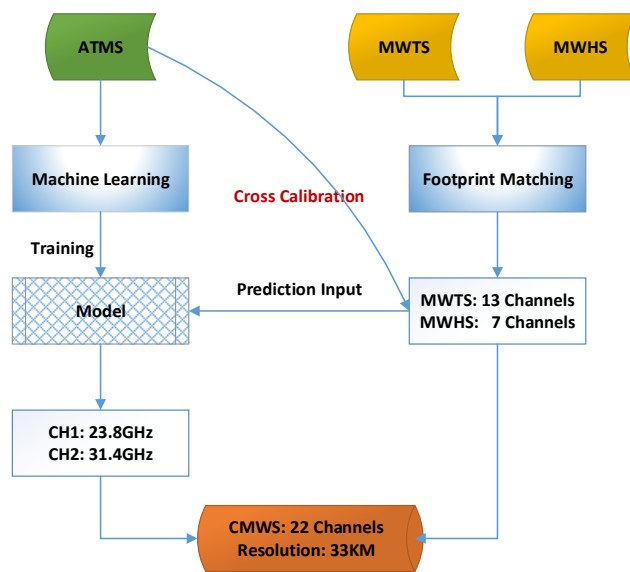

**Figure 1. Flow chart of channel simulation algorithm based on machine learning.**




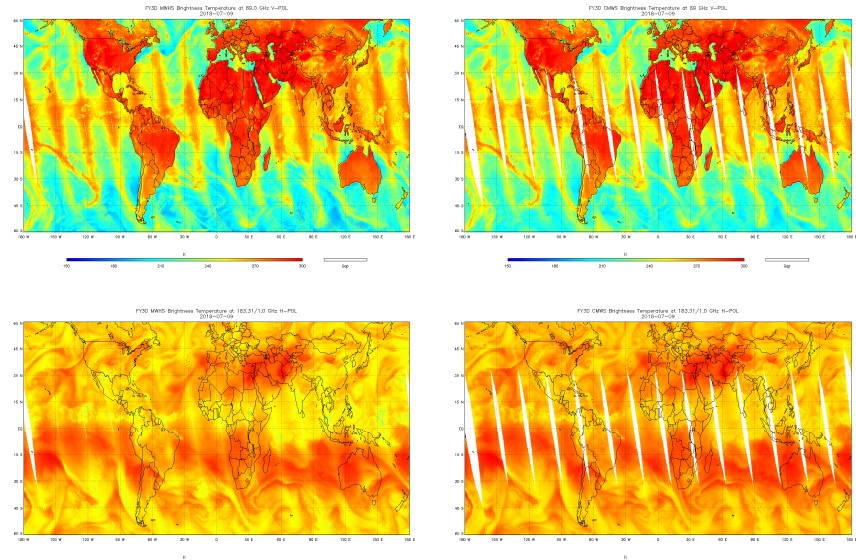

**Figure 2. Footprint matching examples for two typical channels. (left column) the original measurements of MWHS, (right column) the resampled MWHS channels which have the same spatial resolution as MWTS channels.**




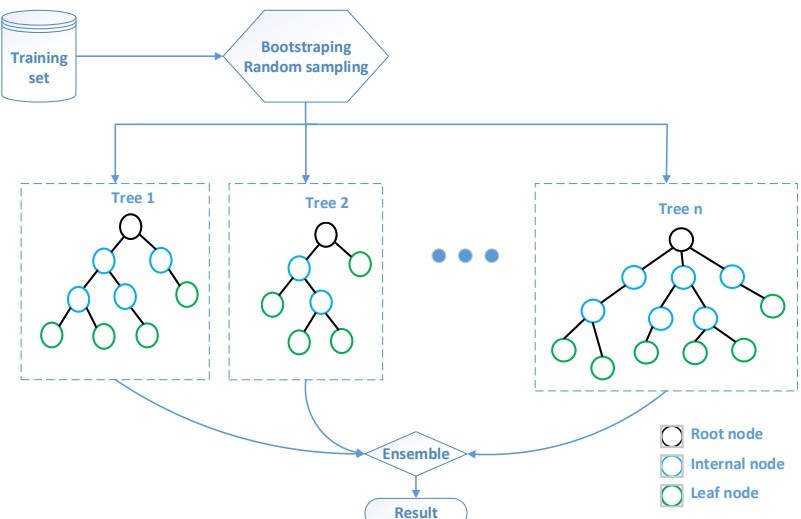

**Figure 3. Schematic diagram of random forest.**





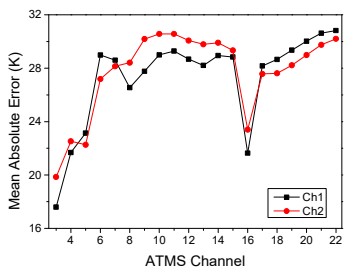 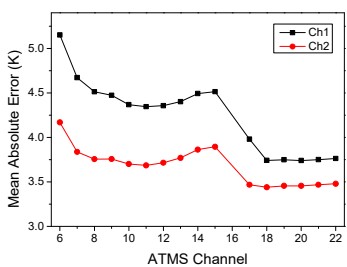

**Figure 4. Simulation errors in different channel combinations. (left panel) Mean absolute error of each single channel, (right panel) mean absolute error when each channel is progressively added to the training model.**





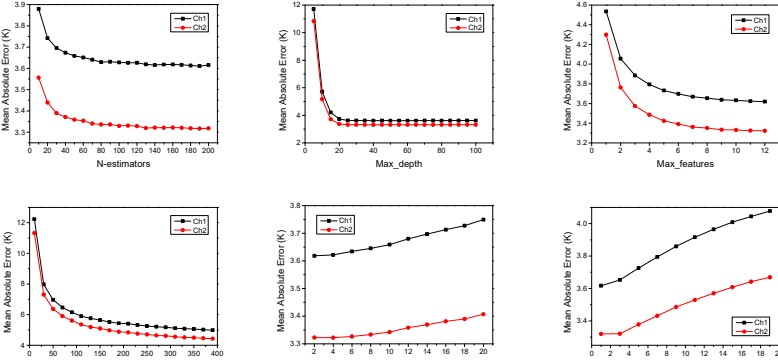

**Figure 5. Simulation errors under different RF model parameters.**

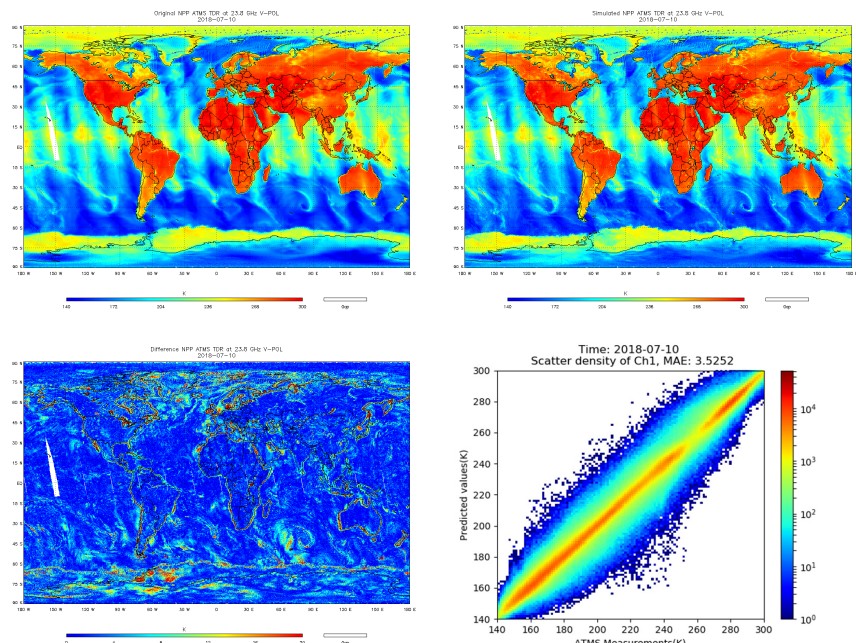

**Figure 6. Accuracy evaluation for Ch1 simulation. (left top) Original measurement, (right top) simulated result, (left bottom) difference result, (right bottom) scatter density.**


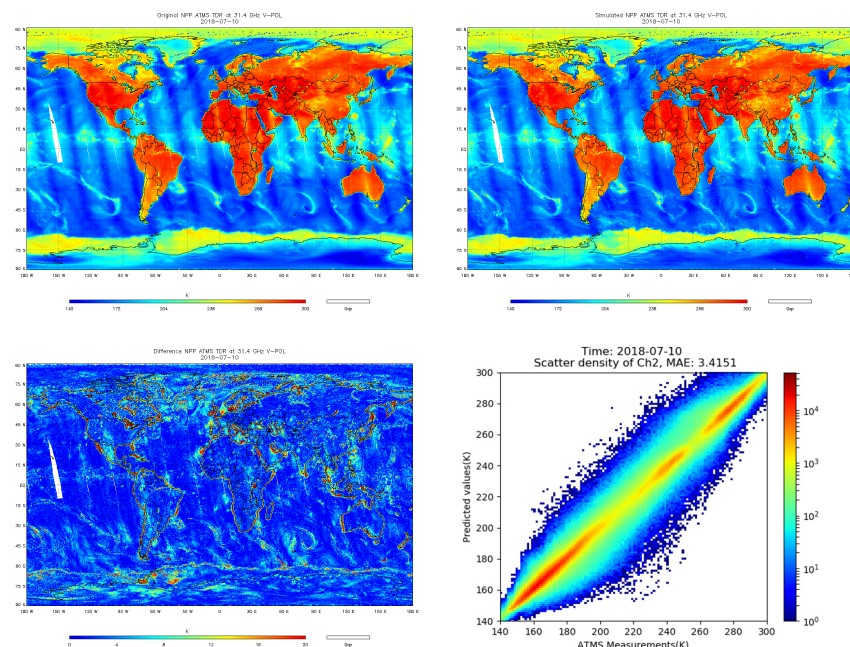

**Figure 7. Accuracy evaluation for Ch2 simulation. (left top) Original measurement, (right top) simulated result, (left bottom) difference result, (right bottom) scatter density.**



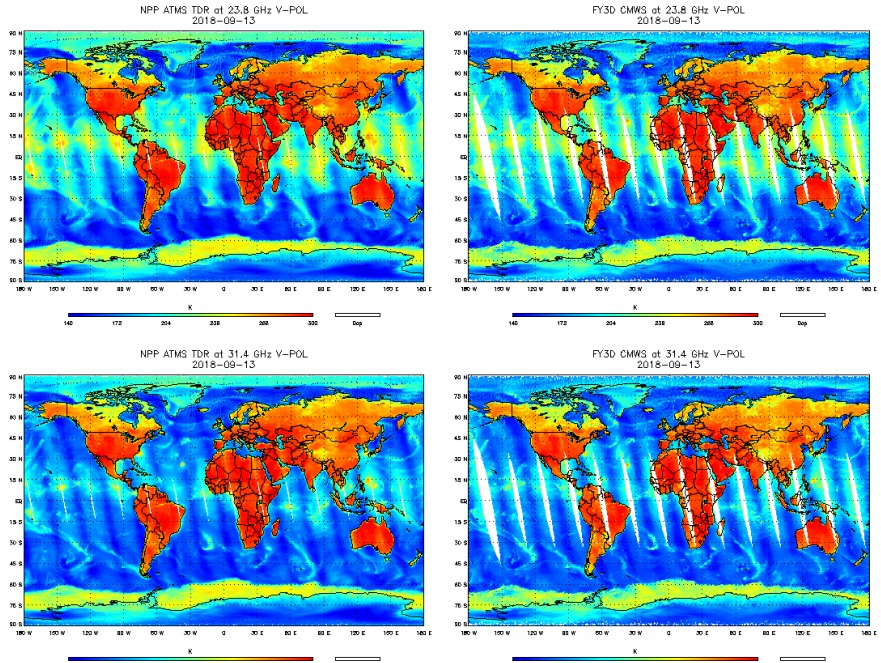

**Figure 8. Compared between the simulation and observation. (left column) the original measurements for Ch1 and Ch2, (right column) Simulation results of two corresponding channels based on FY-3D observation.**




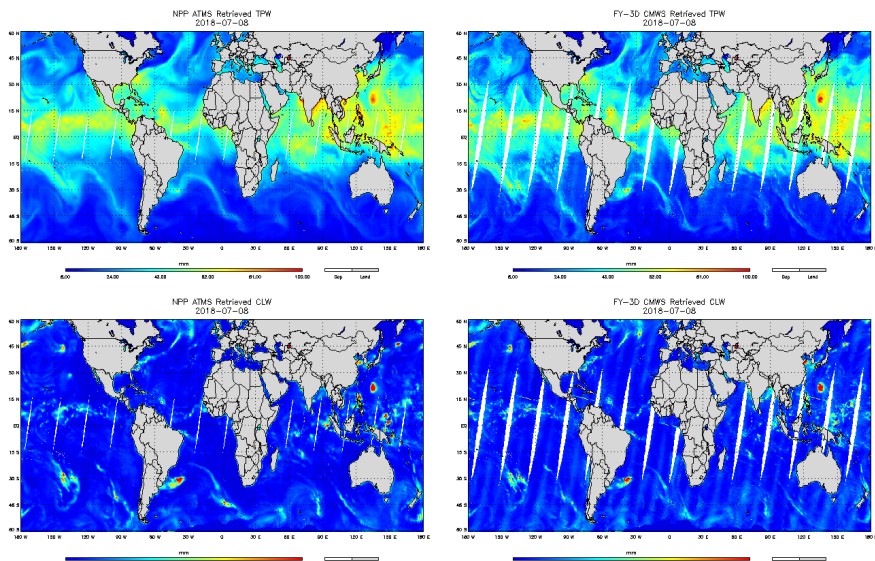

**Figure 9. Comparison of retrieved TPW (top row) and CLW (bottom row) between ATMS (left column) and CMWS (right column) for the descending orbit measurement on July 8, 2018.**



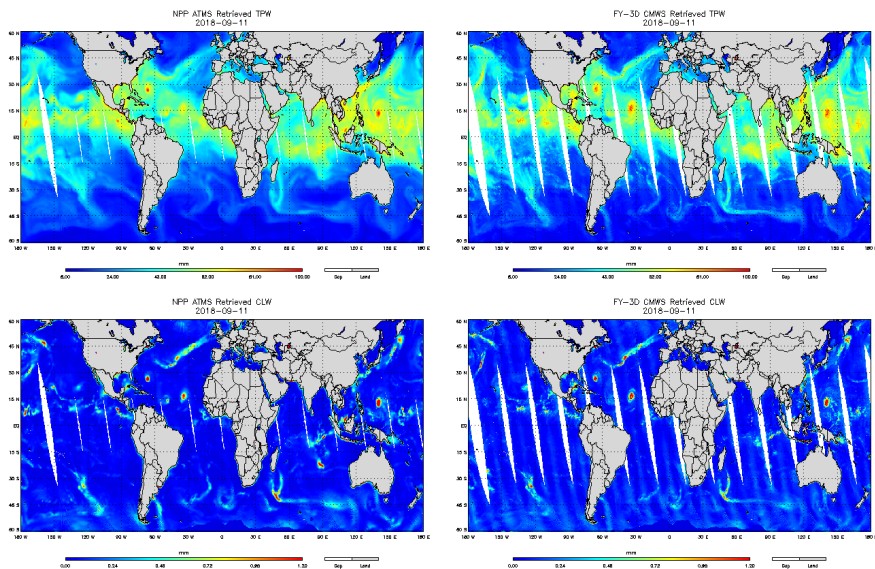

**Figure 10. Comparison of retrieved TPW (top row) and CLW (bottom row) between ATMS (left column) and CMWS (right column) for the ascending orbit measurement on September 11, 2018.**






**Table 1. Slope, intercept and mean absolute error between corrected CMWS (as x values) and ATMS (as y values) brightness temperatures.**


| ATMS Channel Number | Slope | Intercept | Mean Absolute Error (K) |
|---|---|---|---|
| 3 | 1.0124 | -4.6015 | 2.7139 |
| 4 | 0.9665 | 8.8694 | 1.4912 |
| 5 | 0.9980 | 1.9720 | 1.7411 |
| 6 | 1.0021 | 0.0328 | 0.9459 |
| 7 | 0.9944 | 1.7225 | 0.9955 |
| 8 | 0.9533 | 10.8517 | 0.7353 |
| 9 | 0.9978 | -0.7035 | 1.8547 |
| 10 | 1.0258 | -6.7331 | 1.8826 |
| 11 | 1.0201 | -5.2703 | 1.6061 |
| 12 | 1.0163 | -4.2106 | 1.2949 |
| 13 | 1.0277 | -8.5121 | 2.6138 |
| 14 | 0.9827 | 4.3781 | 1.4696 |
| 15 | 1.0255 | -6.6283 | 2.2318 |
| 16 | 0.9374 | 18.6856 | 5.0148 |
| 17 | 0.9392 | 22.8063 | 8.2390 |
| 18 | 0.9747 | 8.0082 | 1.9217 |
| 19 | 1.0314 | -10.4440 | 2.9500 |
| 20 | 0.9592 | 12.2276 | 1.9781 |
| 21 | 0.9311 | 19.6531 | 2.5483 |
| 22 | 0.9588 | 13.2642 | 3.2390 |