# Peer review of "Retrieval of the total precipitable water vapor and cloud liquid water path over ocean from the Feng-Yun 3D microwave temperature and humidity sounders"

_Atmospheric Measurement Techniques, 2019_

## Referee Comment (RC1) · Anonymous Referee #1 · 19 Dec 2019

The work presented in this article is interesting and the authors invested a lot of work to enable a retrieval of cloud liquid water path (CLW) and total precipitable water (TPW) from measurements of MWTS and MWHS aboard FY-3D.

However, although the title of the paper suggested a focus on the retrieval scheme and a thorough analysis and validation of the results, this has been just briefly touched in section 4. The authors are aiming to use existing algorithms to retrieve CLW and TPW, which have been originally developed for another instrument. It is not clear whether the algorithm coefficients have been adapted or newly trained. I have to assume that this is

not the case because it is not presented in the paper and that the original algorithm has been used unchanged. I therefore do not see a substantial new concept or approach to derive CLW and TPW from sounder data.

The verification of the results is very limited. Basically, just two days of retrieved data from two different instruments are compared visually. Unfortunately, a thorough statistical analysis and validation of the results over a sufficient period of time is missing.

The authors concentrate on the simulation of two channels, which are not available on the two instruments but essentially needed to apply the selected algorithms. As these two channels are window channels, I do not expect that the surface emissivity characteristics of these window channels can be recovered from the sounding channels, as these do not see the surface. It is certainly possible to estimate parts of the atmospheric emission of the signal, because it is due to liquid water and water vapour. However, it would then be more appropriate to develop a new retrieval to directly derive CLW and TPW from the existing physical information in the given channel set. The additional step of estimating the surface channels is adding large uncertainties. This could be avoided by a direct retrieval.

Given these major general issues, I do not recommend to accept this paper for publication.

---

## Referee Comment (RC2) · Anonymous Referee #2 · 12 Feb 2020

This study provided a method to retrieve total precipitable water and cloud liquid water path over ocean using observations from FY-3D MWTS and MWHS. It was very useful to extend the usage of FY-3D measurements. However, there are still several questions. 1) The major part of this manuscript was to generate brightness temperature of two frequencies at 23.8 and 31.4Ghz. However, as I know, FY-3D MWRI provides vertical and horizontal observations at 23.8Ghz, why not use these observations directly for the retrieval of TPW and CLW? 2)The manuscript said that the mean absolute errors of the two simulated channels are both between 3 and 4K, I want to know how much errors will be caused in the retrieved TPW and CLW when using the simulated channels. 3)In the cross calibration section, only two days data were used in the cross-calibration between ATMS and CMWS. How the authors guarantee the stability of cross calibration relationship between ATMS and CMWS. 4)There are lots of TPW observations from SuomiNet GPS network and RAOB network on small islands. It will be helpful if these observations were used to validate the retrieved TPW.

---

## Editor Comment (EC1) · Marloes Gutenstein-Penning de Vries (Editor) · 12 Mar 2020

The manuscript presents the considerable work done by the authors to apply the ATMS TPW and CLW retrieval algorithm to the instruments on the Feng-Yun 3D platform. This is an interesting project, as observations from Feng-Yun 3D potentially fill a gap between successive orbits of other satellite-borne microwave instruments. However, the approach adopted by the authors is based on the reconstruction of observations in the window channels at 23.8 and 31.4 GHz, which are not available to MWTS and MWHS. Machine learning is arguably a very powerful tool, yet it cannot re-create information

unique to the missing channels from the observations in other channels. Therefore, the suggestion by both reviewers is to develop a new algorithm based on the strengths of the channels that are available to MWTS and MWHS.

In its current form, the manuscript cannot be accepted for publication in AMT. But I encourage the authors to use the reviewers' recommendations to improve the algorithm, perform extensive validation, and submit an updated version of the paper in due time.
* * *

---

## Author Comment (AC1) · 29 Mar 2020

**Response to Reviewer 1**

We thank the anonymous referee for very thorough and constructive comments. Below are our responses to the comments.

**Reviewer 1**

The work presented in this article is interesting and the authors invested a lot of work to enable a retrieval of cloud liquid water path (CLW) and total precipitable water (TPW) from measurements of MWTS and MWHS aboard FY-3D.
However, although the title of the paper suggested a focus on the retrieval scheme and a thorough analysis and validation of the results, this has been just briefly touched in section 4. The authors are aiming to use existing algorithms to retrieve CLW and TPW, which have been originally developed for another instrument. It is not clear whether the algorithm coefficients have been adapted or newly trained. I have to assume that this is not the case because it is not presented in the paper and that the original algorithm has been used unchanged. I therefore do not see a substantial new concept or approach to derive CLW and TPW from sounder data.

Response: So far, FY-3D's microwave sounding data still fails to effectively enter the assimilation system to serve the weather forecast directly. One of the very important reasons is that it lacks effective quality control methods to obtain observation points under clear sky.

The large-scale TPW and CLW distribution obtained by satellite inversion, in addition to being used for global cloud water resource assessment, another important function is to use as cloud detection criterion for quality control in satellite data assimilation.

The main purpose of this paper is not to propose a new TPW and CLW inversion method, but to extend the application of FY-3D microwave sounding data by using existing TPW and CLW inversion methods so that it can be successfully assimilated into the numerical weather prediction (NWP) system.

The method adopted in this paper is the TPW and CLW inversion algorithm that has been successfully applied to multiple satellite data (AMSU and ATMS), and the inversion results have been widely used as satellite data quality control in NWP systems, such as GSI and GRAPES.

Except for the two missing channels, FY-3D contains all the sensor channels of ATMS, and their channel parameters are basically the same. Through cross calibration, the

CMWS data generated based on FY-3D has very similar observation results to ATMS, so the inversion algorithm suitable for ATMS can be transplanted into CMWS.

The verification of the results is very limited. Basically, just two days of retrieved data from two different instruments are compared visually. Unfortunately, a thorough statistical analysis and validation of the results over a sufficient period of time is missing.

Response: We added some quantitative assessments in the revised manuscript. A total of five days of data were selected as data sources from different months. Since ATMS and CMWS have different field of view (FOV) and satellite transit times, to perform pixel-to-pixel accuracy assessments, we need to collocation all pixels to ensure that the same pixels are evaluated. Successfully matched pixel pairs need to meet the following parameters: imaging time difference is less than 30 minutes, space distance is less than 15KM, satellite height angle difference is less than 10°, and scanning angle difference is less than 20°. After collocate all the ocean pixels from 60°S to 60°N, a total of 180,906 pixels were used for quantitative evaluation.

First, we compared the brightness temperature simulation accuracy of Ch1 and Ch2 when FY-3D observations are used as input in the machine learning model. Figure R1 shows scatter plots between ATMS and CMWS of two corresponding channels. The scatter results for five different dates are shown in (a) to (e), respectively. Subplot (f) represents the total scatter results. Overall, the accuracy and stability of the two simulated channel are satisfactory. According to the five-day observation results from different months, the correlation coefficient of Ch1 is more than 0.9, and the correlation coefficient of Ch2 is also close to 0.9. It should be pointed out that the results of machine learning using FY-3D observations as input will definitely be lower than the accuracy of quantitative evaluation using ATMS measurements as input. This is because the cross calibration between ATMS and FY-3D will inevitably introduce some new errors. Quantitative evaluation results can be found in Table R1. The mean absolute errors of the two channels between ATMS and CMWS are 6.74 and 5.73K, respectively.

[Figure]

[Figure]

[Figure]

Figure R1. Scatter plots for ATMS channels and CMWS channels. (a) June 2, 2018, (b) July 2, 2018, (c) August 2, 2018, (d) September 2, 2018, (e) October 2, 2018, (f) All collocation pixels.

Second, we compared the retrieved TPW and CLW using the same retrieval method for ATMS and CMWS, respectively. Figure R2 shows scatter plots of retrieved TPW and CLW based on ATMS and CMWS, respectively. Also, the scatter results for five different dates are shown in (a) to (e), respectively. Subplot (f) represents the total scatter results. The results of quantitative evaluation show that the correlation coefficients of TPW and CLW between CMWS and ATMS are 0.95 and 0.85, respectively, and the mean absolute errors are 5.14mm and 0.1mm. Moreover, the correlation coefficients and mean absolute errors during the five independent days are very close, which shows that the method proposed in this paper has good stability and robustness (see Table R1).

[Figure]

[Figure]

Figure R2. Scatter plots of retrieved TPW and CLW based on ATMS and CMWS, respectively. (a) June 2, 2018, (b) July 2, 2018, (c) August 2, 2018, (d) September 2, 2018, (e) October 2, 2018, (f) All collocation pixels.

Table R1. Quantitative evaluation results between ATMS and CMWS

| Date (2018) | Matched pixels | Correlation Coefficient | | | | Mean Absolute Error | | | |
|---|---|---|---|---|---|---|---|---|---|
| | | Ch1 | Ch2 | TPW | CLW | Ch1 (K) | Ch2 (K) | TPW (mm) | CLW (mm) |
| June 2 | 54,831 | 0.90 | 0.82 | 0.94 | 0.85 | 7.27 | 8.66 | 5.43 | 0.15 |
| July 2 | 53,322 | 0.94 | 0.90 | 0.95 | 0.89 | 6.75 | 4.4 | 5.34 | 0.08 |
| August 2 | 40,565 | 0.94 | 0.90 | 0.95 | 0.89 | 6.24 | 4.60 | 5.11 | 0.08 |
| September 2 | 22,955 | 0.93 | 0.88 | 0.96 | 0.87 | 6.37 | 4.62 | 4.48 | 0.07 |
| October 2 | 8,936 | 0.88 | 0.88 | 0.89 | 0.86 | 6.61 | 3.77 | 4.03 | 0.07 |
| Total | 180,609 | 0.92 | 0.85 | 0.95 | 0.85 | 6.74 | 5.73 | 5.14 | 0.10 |

The authors concentrate on the simulation of two channels, which are not available on the two instruments but essentially needed to apply the selected algorithms. As these two channels are window channels, I do not expect that the surface emissivity characteristics of these window channels can be recovered from the sounding channels, as these do not see the surface. It is certainly possible to estimate parts of the atmospheric emission of the signal, because it is due to liquid water and water vapour. However, it would then be more appropriate to develop a new retrieval to directly derive CLW and TPW from the existing physical information in the given channel set. The additional step of estimating the surface channels is adding large uncertainties. This could be avoided by a direct retrieval.

Given these major general issues, I do not recommend to accept this paper for publication.

Response: For any FOV, although the observations of each channel are different due to different observation frequencies, there is a certain degree of connection between these observations. This is because the corresponding surface parameters and atmospheric environment at each observation point are exactly the same. It can also be seen from the distribution of the weighting function of all channels in ATMS that although Ch1 and Ch2 are window channels, the weighting distributions of Ch3, Ch4, Ch5 and Ch16 are very similar to those of Ch1 and Ch2, and other channels can also provide certain information

in the lower layer (Figure R3). Therefore, we can establish the relationship between two low-frequency window channels and other channels of ATMS through the machine learning method for all FOVs.

Since MWTS and MWHS contain all channel settings in ATMS except for the two low-frequency window channels, and the weighting function of these channels is also consistent with the corresponding channel in ATMS, we can match FY-3D data to ATMS level by cross calibration, thus realizing the prediction of missing channel values in FY-3D using ATMS training model.

[Figure]

Figure R3 Weighting function of ATMS and FY-3D CMWS

We agree with the reviewer that it is best to retrieve directly from the microwave sounding data of FY-3D, but the information provided by other channels is not enough to invert TPW and CLW. Moreover, the quality control algorithm in the current NWP systems (such as GSI and GRAPES) is mainly based on the two channels (at 23.8 and 31.4GHz). That is, even if there is a newly developed TPW and CLW algorithm based on other channels, the effect of its application to the assimilation system is still uncertain.

Although the inversion of TPW and CLW through the two simulated channels will inevitably introduce some errors, after strict cross calibration and high-precision machine learning training, the correlation coefficients between TPW and CLW retrieved by two simulated channels and those retrieved by ATMS can still reach 0.95 and 0.85. Considering that our inverted TPW and CLW are mainly used for qualitative cloud detection and quality control during satellite data assimilation, the method proposed in this paper still has good application prospects at this stage to ensure that the microwave sounding data of FY-3D can effectively enter the assimilation system. Actually, based on the method proposed in this paper and the advanced radiative transfer modeling system (ARMS), the microwave sounding data of FY-3D has entered the GRAPES forecast system in China in real time.

The next FY-3 satellite FY-3E will soon be launched, in which two window channels (at

23.8 and 31.4GHz) will be added in the new microwave sounder. The simulation method proposed in this paper can also provide a good proxy simulation for FY-3E. After the FengYun satellite has the real observation data of these two window channels, we will carry out the inversion of TPW and CLW use physical inversion method (Weng et al., 2003), which should have higher retrieval accuracy than current statistical inversion method.

Weng, F., Zhao, L., Ferraro, R., Poe, G., Li, X., and Grody, N.: Advanced microwave sounding unit cloud and precipitation algorithms, Radio Sci., 38(4), 8068, doi:10.1029/2002RS002679, 2003.

---

## Author Comment (AC2) · 29 Mar 2020

**Response to Reviewer 2**

We thank the anonymous referee for very thorough and constructive comments. Below are our responses to the comments.

**Reviewer2**

This study provided a method to retrieve total precipitable water and cloud liquid water path over ocean using observations from FY-3D MWTS and MWHS. It was very useful to extend the usage of FY-3D measurements. However, there are still several questions.

1) The major part of this manuscript was to generate brightness temperature of two frequencies at 23.8 and 31.4Ghz. However, as I know, FY-3D MWRI provides vertical and horizontal observations at 23.8Ghz, why not use these observations directly for the retrieval of TPW and CLW?

Response: MWTS, MWHS and MWRI are different microwave loads on FY-3D, and their corresponding spatial resolution and instantaneous field of view are completely different. MWTS and MWHS are mainly used to observe the vertical structure of the atmosphere, while MWRI is mainly used to observe the surface parameters.

The purpose of this paper is to extend the application of FY-3D microwave sounding data by using existing TPW and CLW inversion methods so that it can be successfully assimilated into the numerical weather prediction system. To invert TPW and CLW from microwave measurements, both statistical inversion and physical inversion methods need the information of 23.8 and 31.4GHz channels.

In satellite remote sensing, the observations obtained by different sensors are difficult to be used interchangeably, and they need to undergo strict cross calibration and footprint matching. Although MWRI can provide observations at 23.8GHz, it still lacks the measurements at 31.4GHz. If we want to directly use the observations of MWRI, we need to add an additional cross calibration and footprint matching process. Especially, due to the need to meet the requirements of time and space consistency, the footprint matching between different sensors often results in the loss of many pixels, which will introduce more uncertainty.

2) The manuscript said that the mean absolute errors of the two simulated channels are both between 3 and 4K, I want to know how much errors will be caused in the retrieved TPW and CLW when using the simulated channels.

Response: We added some quantitative assessments in the revised manuscript. A total of five days of data were selected as data sources from different months. Since ATMS and CMWS have different field of view (FOV) and satellite transit times, to perform pixel-to-pixel accuracy assessments, we need to collocation all pixels to ensure that the same pixels are evaluated. Successfully matched pixel pairs need to meet the following parameters: imaging time difference is less than 30 minutes, space distance is less than 15KM, satellite height angle difference is less than 10°, and scanning angle difference is less than 20°. After collocate all the ocean pixels from 60°S to 60°N, a total of 180,906 pixels were used for quantitative evaluation.

First, we compared the brightness temperature simulation accuracy of Ch1 and Ch2 when FY-3D observations are used as input in the machine learning model. Figure R1 shows scatter plots between ATMS and CMWS of two corresponding channels. The scatter results for five different dates are shown in (a) to (e), respectively. Subplot (f) represents the total scatter results. Overall, the accuracy and stability of the two channel simulation are satisfactory. According to the five-day observation results from different months, the correlation coefficient of Ch1 is more than 0.9, and the correlation coefficient of Ch2 is also close to 0.9. It should be pointed out that the results of machine learning using FY-3D observations as input will definitely be lower than the accuracy of quantitative evaluation using ATMS measurements as input. This is because the cross calibration between ATMS and FY-3D will inevitably introduce some new errors. Quantitative evaluation results can be found in Table R1. The mean absolute errors of the two channels between ATMS and CMWS are 6.74 and 5.73K, respectively.

[Figure]

[Figure]

Figure R1. Scatter plots for ATMS channels and CMWS channels. (a) June 2, 2018, (b) July 2, 2018, (c) August 2, 2018, (d) September 2, 2018, (e) October 2, 2018, (f) All collocation pixels.

Second, we compared the retrieved TPW and CLW using the same retrieval method for ATMS and CMWS, respectively. Figure R2 shows scatter plots of retrieved TPW and CLW based on ATMS and CMWS, respectively. Also, the scatter results for five different dates are shown in (a) to (e), respectively. Subplot (f) represents the total scatter results. The results of quantitative evaluation show that the correlation coefficients of TPW and CLW between CMWS and ATMS are 0.95 and 0.85, respectively, and the mean absolute errors are 5.14mm and 0.1mm. Moreover, the correlation coefficients and mean absolute errors during the five independent days are very close, which shows that the method proposed in this paper has good stability and robustness (see Table R1).

[Figure]

Figure R2. Scatter plots of retrieved TPW and CLW based on ATMS and CMWS, respectively. (a) June 2, 2018, (b) July 2, 2018, (c) August 2, 2018, (d) September 2, 2018, (e) October 2, 2018, (f) All collocation pixels.

Table R1. Quantitative evaluation results between ATMS and CMWS

| Date (2018) | Matched pixels | Correlation Coefficient | | | | Mean Absolute Error | | | |
|---|---|---|---|---|---|---|---|---|---|
| | | Ch1 | Ch2 | TPW | CLW | Ch1 (K) | Ch2 (K) | TPW (mm) | CLW (mm) |
| June 2 | 54,831 | 0.90 | 0.82 | 0.94 | 0.85 | 7.27 | 8.66 | 5.43 | 0.15 |
| July 2 | 53,322 | 0.94 | 0.90 | 0.95 | 0.89 | 6.75 | 4.4 | 5.34 | 0.08 |
| August 2 | 40,565 | 0.94 | 0.90 | 0.95 | 0.89 | 6.24 | 4.60 | 5.11 | 0.08 |
| September 2 | 22,955 | 0.93 | 0.88 | 0.96 | 0.87 | 6.37 | 4.62 | 4.48 | 0.07 |
| October 2 | 8,936 | 0.88 | 0.88 | 0.89 | 0.86 | 6.61 | 3.77 | 4.03 | 0.07 |
| Total | 180,609 | 0.92 | 0.85 | 0.95 | 0.85 | 6.74 | 5.73 | 5.14 | 0.10 |

3) In the cross calibration section, only two days data were used in the cross-calibration between ATMS and CMWS. How the authors guarantee the stability of cross calibration relationship between ATMS and CMWS.

Response: The prerequisite for cross calibration is to ensure that the cross-calibration point pairs meet the same observation time and the same observation point. To cross calibrate different sensors, we first need to perform footprint matching. However, Suomi NPP and FY-3D satellites have completely different orbit altitudes and transit times. Most of the time, the same observation point that two sensors can match is very limited.

Fortunately, the sub-satellite trajectories of Suomi NPP and FY-3D satellites are very close each other on February 1-2, 2018, which allows us to use the data of these two days to cross-calibrate ATMS and CMWS. Although we only used two days data, the number of points successfully matched exceeded 100,000 pairs, and they covered the world evenly, including different weather around the world, which is very representative. Once the cross-calibration relationship between different sensors is determined, the mapping relationship between them will not change greatly as long as the sensors do not show severe aging.

4) There are lots of TPW observations from SuomiNet GPS network and RAOB network on small islands. It will be helpful if these observations were used to validate the retrieved TPW.

Response: Thanks for the reviewer's suggestion. Due to the influence of satellite transit time, very few observation points can be matched between the satellite and the ground observations. Considering that the TPW and CLW inversion methods adopted in this paper have been successfully applied in ATMS, we mainly performed comparisons between different satellites to verify the results.

In addition to the qualitative comparison used in the original manuscript, we added some quantitative assessments in the revised manuscript. For specific comparison results, please refer to our reply to the second question of the reviewer.

---

## Author Comment (AC3) · 29 Mar 2020

Dear Editor,

We have thoroughly revised our manuscript and also responded to the comments from two reviewers.

**Associate editor**

The manuscript presents the considerable work done by the authors to apply the ATMS TPW and CLW retrieval algorithm to the instruments on the Feng-Yun 3D platform. This is an interesting project, as observations from Feng-Yun 3D potentially fill a gap between successive orbits of other satellite-borne microwave instruments. However, the approach adopted by the authors is based on the reconstruction of observations in the window channels at 23.8 and 31.4 GHz, which are not available to MWTS and MWHS. Machine learning is arguably a very powerful tool, yet it cannot re-create information unique to the missing channels from the observations in other channels. Therefore, the suggestion by both reviewers is to develop a new algorithm based on the strengths of the channels that are available to MWTS and MWHS.

In its current form, the manuscript cannot be accepted for publication in AMT. But I encourage the authors to use the reviewers' recommendations to improve the algorithm, perform extensive validation, and submit an updated version of the paper in due time.

Response: Thanks for the editor's comment. We have responded to all the questions focused by the reviewers.

From the distribution of the weighting function of all channels in ATMS that although Ch1 and Ch2 are window channels, the weighting distributions of Ch3, Ch4, Ch5 and Ch16 are very similar to those of Ch1 and Ch2, and other channels can also provide certain information in the lower layer (Figure R1). Therefore, we can establish the relationship between two low-frequency window channels and other channels of ATMS through the machine learning method for all FOVs.

Since MWTS and MWHS contain all channel settings in ATMS except for the two low-frequency window channels, and the weighting function of these channels is also consistent with the corresponding channel in ATMS, we can match FY-3D data to ATMS level by cross calibration, thus realizing the prediction of missing channel values in FY-3D using ATMS training model.

We also added some quantitative assessments in the revised manuscript. A total of five days of data were selected as data sources from different months. Quantitative evaluation result for five independent days can be found in Table R1. Overall, the accuracy and

stability of the two channel simulation are satisfactory. The total correlation coefficient of Ch1 is more than 0.9, and the correlation coefficient of Ch2 is also close to 0.9. The MAEs of the two channels between ATMS and CMWS are 6.74 and 5.73K, respectively. Although the simulation errors of the two missing channels still seem to be a little large, the accuracy and stability of the two channel simulation are still satisfactory, especially the simulation accuracy is sufficient to meet the quality control requirements of satellite data assimilation. After strict cross calibration and high-precision machine learning training, the correlation coefficients between TPW and CLW retrieved by two simulated channels and those retrieved by ATMS can still reach 0.95 and 0.85.

[Figure]

Figure R1 Weighting function of ATMS and FY-3D CMWS

Table R1. Quantitative evaluation results between ATMS and CMWS

| Date (2018) | Matched pixels | Correlation Coefficient | | | | Mean Absolute Error | | | |
|---|---|---|---|---|---|---|---|---|---|
| | | Ch1 | Ch2 | TPW | CLW | Ch1 (K) | Ch2 (K) | TPW (mm) | CLW (mm) |
| June 2 | 54,831 | 0.90 | 0.82 | 0.94 | 0.85 | 7.27 | 8.66 | 5.43 | 0.15 |
| July 2 | 53,322 | 0.94 | 0.90 | 0.95 | 0.89 | 6.75 | 4.4 | 5.34 | 0.08 |
| August 2 | 40,565 | 0.94 | 0.90 | 0.95 | 0.89 | 6.24 | 4.60 | 5.11 | 0.08 |
| September 2 | 22,955 | 0.93 | 0.88 | 0.96 | 0.87 | 6.37 | 4.62 | 4.48 | 0.07 |
| October 2 | 8,936 | 0.88 | 0.88 | 0.89 | 0.86 | 6.61 | 3.77 | 4.03 | 0.07 |
| Total | 180,609 | 0.92 | 0.85 | 0.95 | 0.85 | 6.74 | 5.73 | 5.14 | 0.10 |